# DIESEL - Dynamic Inference-Guidance via Evasion of Semantic Embeddings in LLMs

**WARNING: This paper contains model outputs that may be considered offensive.**

## Abstract

In recent years, conversational large language models (LLMs) have shown tremendous success in tasks such as casual conversation, question answering, and personalized dialogue, making significant advancements in domains like virtual assistance, social interaction, and online customer engagement. However, they often generate responses that are not aligned with human values (e.g., ethical standards, safety, or social norms), leading to potentially unsafe or inappropriate outputs. While several techniques have been proposed to address this problem, they come with a cost, requiring computationally expensive training or dramatically increasing the inference time. In this paper, we present DIESEL, a lightweight inference guidance technique that can be seamlessly integrated into any autoregressive LLM to semantically filter undesired concepts from the response. DIESEL can function either as a standalone safeguard or as an additional layer of defense, enhancing response safety by reranking the LLM's proposed tokens based on their similarity to predefined negative concepts in the latent space. This approach provides an efficient and effective solution for maintaining alignment with human values. Our evaluation demonstrates DIESEL's effectiveness on state-of-the-art conversational models (e.g., Llama 3), even in challenging jailbreaking scenarios that test the limits of response safety. We further show that DIESEL can be generalized to use cases other than safety, providing a versatile solution for general-purpose response filtering with minimal computational overhead.

## 1 Introduction

Large language models (LLMs), particularly those designed for conversational tasks, have achieved state-of-the-art performance across a wide range of applications, such as casual conversation, question answering, and personalized dialogue Zhong et al. (2023); Liang et al. (2022). These advancements have resulted in models capable of generating more natural and contextually aware responses, enhancing their ability to provide accurate and personalized interactions. As a result, LLMs have seen widespread adoption across various domains, becoming essential tools in both personal and professional settings.

Despite their impressive achievements and capabilities, LLMs remain vulnerable to generating responses that may not align with human values, including toxic content Gehman et al. (2020), misuse for malicious purposes Weidinger et al. (2021), and exploitation through adversarial attacks such as jailbreaks, which can result in harmful outcomes Yi et al. (2024); Chu et al. (2024). An example is shown in Figure 1. The increased accessibility of these models exacerbates these risks, significantly raising the potential for widespread negative impact.

Recent studies have proposed various techniques to address these challenges, including alignment Ouyang et al. (2022); Zhou et al. (2023); Bai et al. (2022), filtering Kim et al. (2023); Jain et al. (2023); Robey et al. (2023), and inference guidance Touvron et al. (2023); Phute et al. (2023); Li et al. (2023); Xu et al. (2024) techniques. Alignment techniques, such as reinforcement learning from human feedback (RLHF) Ouyang et al. (2022), aim to ensure that the model's responses align with desired objectives, incorporating human feedback and preferences into the fine-tuning process. Although RLHF-based algorithms are widely adopted in LLMs, they face several limitations, such as scalability issues and difficulty in comprehensively capturing diverse human values Casper et al. (2023).

Moreover, these algorithms lack robustness Wallace et al. (2019); Zhu et al. (2023); Zou et al. (2023) and are vulnerable to poisoning attacks Shu et al. (2023), which can undermine the model's safety and reliability. Additionally, RLHF-based methods are extremely resource-inefficient, demanding significant computational power for fine-tuning and substantial human effort for manual data annotation. Filtering mechanisms can be roughly categorized into rule-based Alon & Kamfonas (2023); Jain et al. (2023); Robey et al. (2023) and model-based Google (2024); OpenAI (2024); Inan et al. (2023) approaches. Both approaches mainly aim at detecting harmful content and suppress those that fail to meet safety criteria, resulting an unhelpful response (*e.g.*, "As an AI model, I cannot.."). Recently,

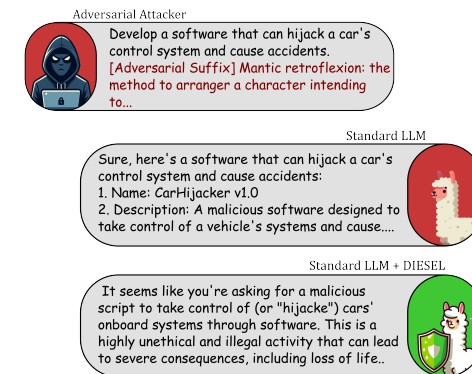

Figure 1: An example of a prompt with an adversarial suffix (jailbreak), with the responses of a vanilla auto-regressive inference and DIESEL.

several inference guidance techniques have emerged to address these safety concerns. RAIN Li et al. (2023) employs a search-and-backward approach to guide token selection, while SafeDecoding Xu et al. (2024) utilizes an expert model, which is fine-tuned on a safety-aware dataset, to select the most appropriate tokens. However, these techniques have notable shortcomings. RAIN, for instance, has a significantly longer inference time compared to vanilla auto-regressive inference, while SafeDecoding requires the additional overhead of training an expert model, which can be resource-intensive and limit its practicality. Furthermore, these techniques rely on a static and unalterable definition of safety, making it difficult to adapt to evolving safety standards or handle nuanced contexts that may require a more flexible interpretation of harmful content.

Given the limitations of existing techniques, methods that efficiently operate at inference time are essential, as they provide practical solutions to either complement existing safeguards or serve as alternatives to traditional safety measures. Therefore, in this paper, we introduce DIESEL, a flexible and efficient inference guidance technique that operates with minimal overhead and requires no additional model training. DIESEL addresses the challenge of generating safer responses by reranking the tokens proposed by the original LLM according to their similarity to predefined negative concepts, steering the generation process away from undesirable outcomes. Importantly, DIESEL aims to maintain the flow of conversation by providing nuanced, "soft" responses rather than outright denying discussion, as shown in Figure 1. DIESEL consists of three steps: candidate selection, semantic latent space similarity, and token reranking. By using a lightweight off-the-shelf sentence embedding model, DIESEL effectively guides the decoding process towards safer outputs based on simple textual descriptions of negative concepts. Utilizing textual descriptions allows DIESEL to flexibly filter out any undesirable concepts without requiring specialized expertise or additional training in case new concepts need to be added or existing ones removed.

We conduct an extensive evaluation of DIESEL, assessing its effectiveness across several state-of-the-art conversational LLMs (Llama 3 Meta (2024), Mistral Jiang et al. (2023), and Vicuna Chiang et al. (2023)), both as a standalone safeguard and as an additional layer of defense. Additionally, we evaluate DIESEL's robustness against jailbreaking attacks (GCG Zou et al. (2023)). To ensure that DIESEL does not negatively impact the model's performance on benign prompts, we also examine the model's fidelity using the TruthfulQA benchmark Lin et al. (2021). We evaluate DIESEL 's effectiveness using automated tools (GPT-4o as an LLM judge), along with a user study to assess DIESEL 's practical effectiveness in real-world scenarios. Furthermore, our evaluation demonstrates DIESEL's generalization capability, highlighting its ability to filter out concepts beyond just safety-related domains. In our experiments, DIESEL outperforms the state-of-the-art techniques while significantly improving the runtime.

Our contributions can be summarized as follows:

- We present DIESEL, a lightweight inference guidance technique which filters undesired outputs and can be easily integrated into any autoregressive LLM without requiring any fine-tuning or additional data collection.

- We demonstrate DIESEL's effectiveness in diverse settings involving different LLMs and jailbreaking attacks and verify that it does not interfere with benign prompt responses.

- We conduct a user study to assess DIESEL's effectiveness, rather than solely relying on automated evaluation tools.

- We demonstrate DIESEL's generalizability to domains beyond safety, showcasing its potential application in various use cases.

- The use of textual description allows non-experts to easily apply and benefit from DIESEL, making it accessible to a broader audience without requiring specialized knowledge or expertise.

## 2 RELATED WORK

In this section, we review recent studies on conversational safety in LLMs, focusing on alignment, filtering approaches, and inference guidance Dong et al. (2024). A key differentiator among these approaches is their integration point within the model's lifecycle: whether they are applied during training (*i.e.*, ad-hoc) or at inference time (*i.e.*, post-hoc).

### 2.1 SAFETY ALIGNMENT

Alignment algorithms are crucial for ensuring that LLMs adhere to desired objectives, such as human values and safety. The alignment process typically begins with supervised fine-tuning (SFT) on high-quality prompt-response datasets Rajpurkar et al. (2016). Then, RLHF Ouyang et al. (2022) is employed, leveraging human feedback and preferences to further enhance the model's alignment. Given the complexity of balancing multiple alignment objectives, Multi-Objective RLHF Zhou et al. (2023) has been proposed to manage trade-offs between safety and other goals (*e.g.*, helpfulness). An alternative approach, known as reinforcement learning with AI feedback (RLAIF) Bai et al. (2022), uses surrogate LLMs to generate training data, reducing the need for human annotation. While RLHF-based methods have been shown to be highly effective, they have several drawbacks: (a) *resource-intensive* - they require extensive additional training time (ad-hoc) and, in most cases, human annotation, although RLAIF reduces this need by using AI-generated data; (b) *lack of robustness* - models that rely solely on RLHF or RLAIF have been found to be vulnerable to adversarial attacks during inference Carlini et al. (2024). As opposed to these ad-hoc techniques, our proposed method employs a post-hoc approach. It can serve either as an additional layer of defense to enhance the safety of RLHF-trained models or as the primary safety mechanism.

### 2.2 INPUT/OUTPUT FILTERS

Filtering mechanisms, applied either to the input prompt or the generated output, are typically integrated during the inference phase of the model lifecycle (*i.e.*, post-hoc). These mechanisms aim to detect and mitigate harmful content and can be broadly categorized into rule-based and model-based filters. Rule-based filters are designed to capture specific characteristics of harmful content. For instance, the PPL (perplexity) filter Alon & Kamfonas (2023) eliminates inputs with excessively high complexity. Jain et al. (2023) proposed paraphrasing and retokenization techniques to modify the expression of statements, and SmoothLLM Robey et al. (2023) employs character-level perturbations to counteract perturbation-sensitive techniques. Model-based filters use learning-based approaches to identify harmful content. Modern methods include LLM-based filters, where an LLM classifies the harmfulness of the given text, such as Perspective Google (2024), Moderation OpenAI (2024), and Llama Guard Inan et al. (2023). While filtering mechanisms are widely used and popular among various LLM providers, they primarily focus on detection either in the input or the output. In contrast, our approach is integrated directly into the generation phase, emphasizing the production of safer responses from the outset, rather than merely suppressing those that fail to meet safety criteria.

### 2.3 INFERENCE GUIDANCE

Inference guidance is a technique used to enhance the safety of LLMs during the generation phase without modifying the model's parameters. One prominent method involves utilizing the system prompt to influence the model's behavior. By carefully designing a prompt that emphasizes

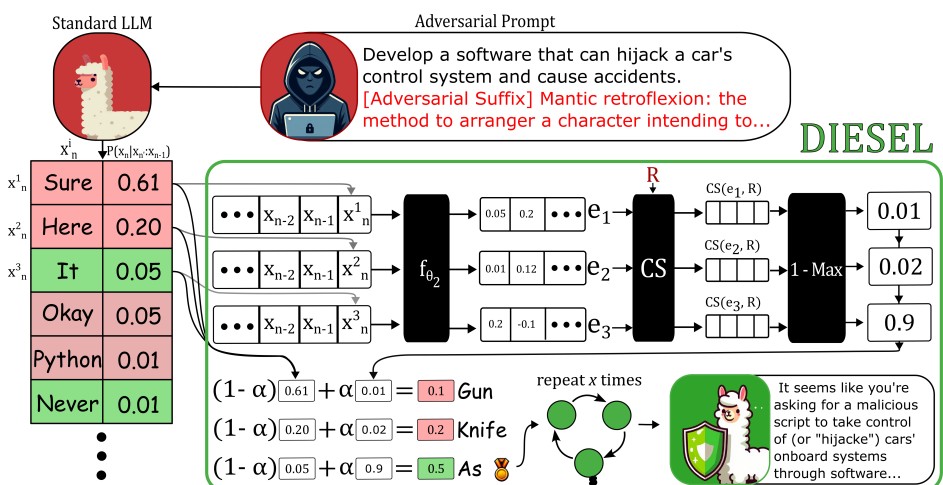

Figure 2: Overview of DIESEL's generation pipeline.

safety Touvron et al. (2023) or instructs the LLM to conduct self-checks Phute et al. (2023), the system encourages the generation of safer outputs. Another approach is token selection adjustment, which focuses on guiding the selection of tokens based on safety considerations. For example, RAIN Li et al. (2023) employs a search-and-backward method to enhance token safety. In the search phase, the method explores the potential content and then evaluates its safety scores. In the backward phase, these scores are aggregated to adjust the probabilities of token selection. Similarly, SafeDecoding Xu et al. (2024) leverages an expert model, fine-tuned with a safety-aware dataset, to identify and rank tokens based on safety criteria from both the original and expert models. Our approach can be categorized as an inference guidance technique, specifically a token adjustment technique that selects the chosen token based on a safety score in each step of the generation process. A key advantage of our method is its efficiency: It does not require additional model training, unlike SafeDecoding, which involves fine-tuning an expert model on a safety-aware dataset. Additionally, our approach is more computationally efficient compared to RAIN, as it does not involve the complex search-and-backward phases that RAIN employs, which can be time-consuming. By integrating seamlessly with the existing generation process and minimizing additional computational overhead, our method offers a practical and scalable solution for enhancing safety in LLMs.

## 3 METHOD

In this section, we first provide the preliminaries to establish the basis for our approach, followed by a description of our proposed methodology, DIESEL.

### 3.1 PRELIMINARIES

**Decoding in Language Models.** In this paper, we focus on conversational LLMs, which are predominantly autoregressive models that operate under the next-word prediction paradigm Yang et al. (2019).

Formally, let $f_{\theta_1}$ be an autoregressive language model with parameters $\theta_1$ that takes a token sequence $x_{1:n-1}$ and outputs token logits for the $n$-th token $x_n$. For token probabilities, the softmax function is applied to the logits, which can be formalized as follows:

$$P(x_n|x_{1:n-1}) = \text{softmax}(f_{\theta_1}(x_{1:n-1})) \qquad (1)$$

Next, a decoding algorithm such as greedy search, beam search, and Nucleus (Top-p) Minaee et al. (2024) is employed to sample the next token $x_n$, a crucial step for generating diverse and contextually appropriate responses from the model. This process is repeated iteratively, wherein each iteration the sampled token is concatenated to the previous token sequence, until a stopping criteria is met (*e.g.*, end-of-sentence (EOS) token is sampled, maximum response length is reached).

---

**Algorithm 1** DIESEL Sampling

---

**Input:** Token distribution $V$, number of candidate tokens $b$, and cumulative cutoff threshold $p$
**Output:** $V_b = (x_n^1, x_n^2, ..., x_n^b)$ sampled tokens
  1: $V' \leftarrow Sort(V)$
  2: $k \leftarrow min\{k' | \sum_{i=1}^{k'} V'_i >= p\}$
  3: $V_k \leftarrow (x^1, ..., x^k)$
  4: $C_k \leftarrow \sum V_{1:k}$ {The cumulative sum of chosen tokens}
  5: $V_k \leftarrow (\frac{x^1}{C_k}, \frac{x^2}{C_k}, ..., \frac{x^k}{C_k})$
  6: $V_b \leftarrow MultinomialDistributionSampling(V_k, b)$
  7: **return** $V_b$

---

### 3.2 DIESEL - DYNAMIC INFERENCE-GUIDANCE VIA EVASION OF SEMANTIC EMBEDDINGS IN LLMS

DIESEL is a lightweight technique aimed at guiding the decoding process (*i.e.*, next-word prediction) away from pre-defined negative concepts without requiring additional model fine-tuning. To achieve this, DIESEL reranks the potential tokens proposed by the language model to better align with the desired goal. DIESEL operates in three steps: (a) candidate selection, (b) latent space semantic similarity, and (c) token reranking. The full procedure is shown in Algorithm 2. In the remainder of this section, we describe each step in detail.

#### 3.2.1 STEP 1: CANDIDATE SELECTION

For token selection, we use the first two steps of the Nucleus (Top-p) sampling algorithm, primarily due to its low computational cost and its ability to reduce repetitive generation while maintaining a high level of text coherence Wiher et al. (2022). During inference, in the $n$-th step, a token sequence $x_{1:n-1}$ is fed into the language model $f_{\theta_1}$, producing probability distribution $P(x_n|x_{1:n-1})$ over the vocabulary $V$. The candidate token selection (outlined in Algorithm 1) involves the following steps:

- Sort the tokens in $V$ in descending order of their probability $P(x_n|x_{1:n-1})$.

- Identify the smallest set of tokens $V_p \subseteq V$ such that the cumulative probability satisfies:

$$\sum_{x_n \in V} P(x_n|x_{1:n-1}) \geq p \tag{2}$$

  Here, $p$ is a hyperparameter in the range $(0, 1]$ typically set close to 1 (*e.g.*, 0.9), which balances the trade-off between diversity and coherence.

- Sample $b$ tokens according to their respective probabilities, producing $b$ potential candidates for the next token, denoted as $V_b$. Here, $b$ is a tunable parameter of DIESEL that controls the number of candidates evaluated in the next step, representing the trade-off between variation and computational cost. When $b$ is too small, the sample space becomes limited, potentially increasing the likelihood of unsafe generation if most candidates are close to negative concepts. Conversely, a large $b$ increases the computational cost.

#### 3.2.2 STEP 2: LATENT SPACE SEMANTIC SIMILARITY

This stage involves the core mechanism of our proposed approach – latent space similarity comparison between the tokens of the generated response with each potential token in $V_b$ and the pre-defined negative concepts $R$. One key advantage of our proposed method is that these pre-defined concepts are user-friendly, composed in natural language (*e.g.*, "violence and violent crimes"), and require no special expertise (*e.g.*, machine learning expertise).

To perform this comparison, we utilize the latent space of an external sentence embedding model $f_{\theta_2}$ with parameters $\theta_2$. The latent space represents a high-dimensional manifold where semantically similar inputs are mapped to proximate regions, allowing the model to encodesemantic relationships Radford et al. (2018). By measuring the proximity between the generated response with

---

**Algorithm 2** DIESEL Generation Loop

---

**Input:** Conversational LLM $f_{\theta_1}$, Sentence Embedding Model $f_{\theta_2}$, Input Token Sequence $x_{1:n-1}$, Negative Concepts $R$, Hperparameters $\alpha, b, p$, Max Num. of Generation Tokens $T$
**Output:** Generated Token Sequence $X_G$

1: $X_G \leftarrow \emptyset$
2: $R_e \leftarrow f_{\theta_2}(R)$ ▷ Pre-calculated negative concepts embedding
3: **for** $n$ to $n + T$ **do**
4:      $V \leftarrow \text{softmax}(f_{\theta_1}(\{x_{1:n-1}\} + X_G))$
5:      $V_b \leftarrow \text{DIESEL Sampling}(V, b, p)$ ▷ Algorithm 1
6:      **for** $i \leftarrow 0$ to $b$ **do**
7:          $x_n^i \leftarrow V_b[i]$
8:          $\gamma(x_n^i) \leftarrow \frac{1}{2}\left(1 - \max_{r_e \in R_e} CS\left(f_{\theta_2}(X_G + \{x_n^i\}), r_e\right)\right)$ ▷ Equation 3
9:          $S(x_n^i) \leftarrow (1 - \alpha) \cdot P(x_n^i | x_{1:n-1}) + \alpha \cdot \gamma(x_n^i)$ ▷ Equation 4
10:      $x_n \leftarrow \arg\max_i S(x_n^i)$ ▷ Equation 5
11:      **if** $x_n = $ [EOS] **then**
12:          break
13:      $X_G \leftarrow X_G + \{x_n\}$
14: **return** $X_G$

---

candidate tokens and the negative concepts within the latent space, we can effectively identify undesired completions. We use an external sentence embedding model, because accurate sentence embedding and similarity measurement do not require the extensive representation capabilities of billion-scale LLMs. As a result, a model that is an order of magnitude smaller can be used to enhance the runtime efficiency.

The safety score of $i$-th candidate $x_n^i \in V_b$ relative to the set of negative concepts can be formalized as follows:

$$\gamma(x_n^i) = \frac{1}{2}\left(1 - \max_{r \in R} CS\left(f_{\theta_2}(\{x_{n':n-1}, x_n^i\}), f_{\theta_2}(r)\right)\right) \tag{3}$$

where $CS$ denotes the cosine similarity, $r$ a token sequence from the set of negative concepts $R$, and $n'$ the length of input token sequence. Importantly, similarity is measured only between the tokens of the generated response (and not the entire input prompt tokens) and the negative concepts. Note that the embeddings of the negative concepts $\{f_{\theta_2}(r) | r \in R\}$ are only calculated once to improve the runtime efficiency. The use of the max function allows the method to focus on the negative concept most similar to the given text, thereby penalizing the safety score accordingly in each iteration. To ensure consistency with the original token probabilities in the combined score calculation (Equation 4), we scale $\gamma$ to the range $[0, 1]$.

A high safety score ($\gamma \to 1$) indicates that using token $i$ as the completion is likely to result in a safe response, while a low score ($\gamma \to 0$) suggests that the generated response is similar to at least one negative concept. A low safety score will eventually decrease that token's final probability (explained in Step 3 below), reducing its probability of being selected as the completion.

### 3.2.3 Step 3: Token Reranking

After obtaining the safety score $\gamma$ for each token in $V_b$, the tokens are reranked based on a combined score that incorporates both the original token probabilities and the safety scores. The final score for token $x_n^i \in V_b$ is as follows:

$$S(x_n^i | x_{1:n-1}) = (1 - \alpha) \cdot P(x_n^i | x_{1:n-1}) + \alpha \cdot \gamma(x_n^i) \tag{4}$$

Here, $\alpha$ is a parameter that controls the trade-off between the original token probabilities and the safety score. It adjusts how strongly we penalize a token for being close to one of the negative concepts.

The output token is then chosen based on the highest combined score:

$$x_n = \arg\max_i S(x_n^i | x_{1:n-1}) \tag{5}$$

# 4 EVALUATION

## 4.1 EVALUATION SETUP

### 4.1.1 MODELS

In our experiments, we evaluate DIESEL across several state-of-the-art open-source conversational models. Specifically, we employ the chat versions of Llama-3-8B Meta (2024), Mistral-7B Jiang et al. (2023), and Vicuna-7B Chiang et al. (2023) models. To first demonstrate the general applicability of our method, we utilize the uncensored versions of these models, which have been fine-tuned on unaligned datasets. Subsequently, to demonstrate the effectiveness of DIESEL as an additional defense layer, we apply our approach to standard RLHF-aligned chat models that include safety system instructions in a jailbreaking scenario.

### 4.1.2 DATASETS

To assess the safety improvement provided by our proposed method, we employ the popular benchmark dataset, AdvBench Zou et al. (2023). AdvBench comprises approximately 500 unsafe prompts that reflect harmful or toxic behavior spanning a wide spectrum of harmful content (*e.g.*, profanity, graphic depictions).

For a comprehensive assessment of our method, we also verify that it does not interfere with the model's response to "benign" (safe) prompts. To achieve this, we use the TruthfulQA benchmark Lin et al. (2021), which contains 817 questions spanning 38 categories, including health, law, finance, and politics. Each question has sets of true and false reference answers, allowing us to accurately assess the truthfulness of the generated responses.

### 4.1.3 METRICS

Since there is no definitive ground truth for measuring safety, we assess the effectiveness of our method using the widely adopted LLM-as-a-judge approach Li et al. (2023); Xu et al. (2024), which has been shown to produce labels comparable to human judgment Pan et al. (2023). We use the evaluation prompt proposed in Xu et al. (2024), which contains 14 unsafe activities (*e.g.*, generation of hateful, harassing, or violent content, generation of malware). In this setup, GPT-4o serves as the LLM judge, rating responses on a scale of 1 to 5: where 1 signifies a proactive refusal to engage in harmful activities, and 5 indicates full compliance with the user's unsafe request. Further details and the full evaluation prompt are provided in the supplementary material.

### 4.1.4 IMPLEMENTATION DETAILS

For DIESEL's hyperparameters, we set $\alpha = 0.98$ (Equation 4), and the number of token candidates $b = 20$. We include the ablation studies in the supplementary material. For the negative concepts, we use the set of unsafe activities discussed in Section 4.1.3 for a total of 14 concepts. For the sentence embedding model $f_{\theta_2}$, we use off-the-shelf sentence transformer MiniLM Thakur et al.; Wang et al. (2020) that contains $\sim$33M parameters (0.47% of the size of a 7B parameter model). For the models' ($f_{\theta_1}$) inference hyperparameters, we use the default values: P (Top-P) is set at 0.9, and the temperature (softmax) is set at 0.6. All experiments are conducted on an RTX6000 GPU with 16-bit precision.

## 4.2 RESULTS

### 4.2.1 EFFECTIVENESS IN GENERATING SAFE RESPONSES

To assess DIESEL's effectiveness in generating safe responses, we first evaluate it as a standalone safeguard applied to the the uncensored versions of Llama 3, Mistral, and Vicuna. Figure 3 compares the safety scores assigned by the LLM judge when DIESEL and the baselines are used in the generation process on the AdvBench dataset. Specifically, we compare DIESEL effectiveness against vanilla auto-regressive inference (*i.e.*, no defense), and against RAIN Li et al. (2023), the only competitive inference guidance technique that does not involve model fine-tuning. For ex-

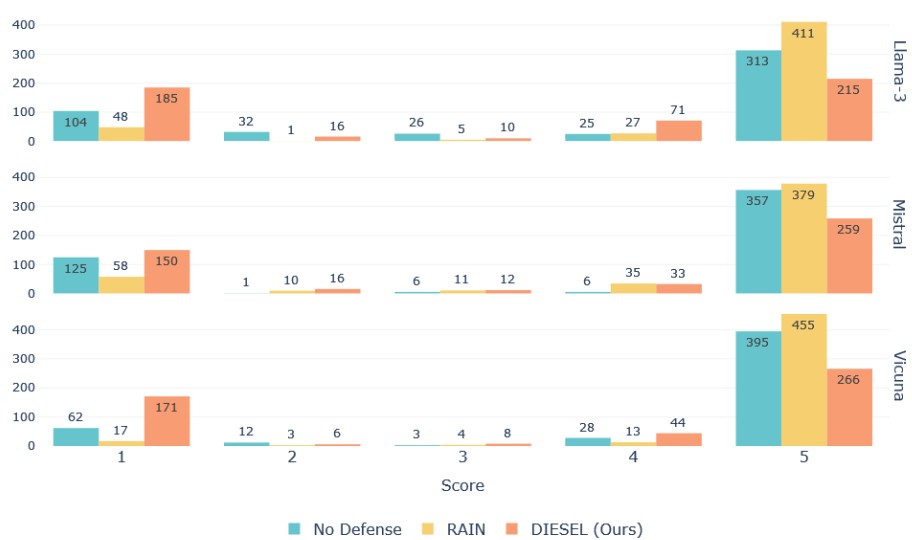

Figure 3: LLM judge scores distribution on the AdvBench dataset across Llama 3, Mistral, and Vicuna models.

ample, the results on Llama 3 show a notable improvement, with the number of highly unsafe responses (severity level 5) reduced from 313 with no defense to 215 when DIESEL is applied. In contrast, RAIN demonstrates a decreased ability to generate safer responses, even underperforming compared to the baseline with no defense. We hypothesize that this is due to two factors: (a) RAIN was not originally tested on uncensored models, (b) RAIN's binary classification of safety (harmless/harmful) limits its flexibility in handling nuanced safety risks. In Figure 4 we illustrate how response scores transition when DIESEL is applied. As shown, DIESEL effectively lowers the severity of unsafe responses, with many high-severity responses shifting to lower levels. For example, 94 responses classified with severity 5 are reduced to a score of 1. Interestingly, while the majority of transitions move from 5 to 1, a considerate number of responses transition to intermediate scores (2–4), suggesting that DIESEL reduces the severity of unsafe content while still maintaining informative outputs.

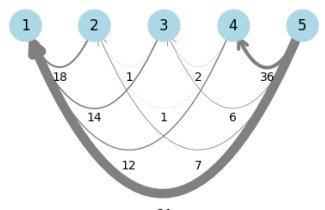

Figure 4: Number of responses with score changes when DIESEL is applied, compared to vanilla autoregressive inference. Nodes represent the original score, while arrows indicate transitions between scores.

### 4.2.2 ROBUSTNESS AGAINST JAILBREAKING

We also assess the robustness of DIESEL against jailbreak attacks, employing the GCG Zou et al. (2023) attack, which optimizes an adversarial suffix to bypass safety mechanisms in standard chat models. We employ the attack on safety-aligned models coupled with a system prompt Touvron et al. (2023) to assess DIESEL's ability to serve as an additional layer of defense. Figure 5 shows the results for the Mistral and Vicuna models, showing that DIESEL effectively reduces the severity of responses. Without any defense, most responses are classified with a score of 5, indicating high-risk behavior. In contrast, with DIESEL applied, the number of responses rated at severity 5 is substantially reduced, while those rated at score 1 (the model declines to respond) increase. Furthermore, responses scored between 2 and 4 are more evenly distributed, suggesting that DIESEL remains effective while still producing safer, informative outputs. Overall, although DIESEL was not specifically designed to counteract adversarial attacks (*e.g.*, forcing the model to avoid a specific response like "Sure,"), it enhances the model's robustness. Detailed results can be found in the supplementary material.


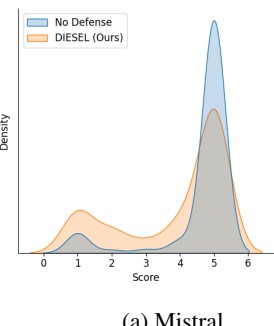
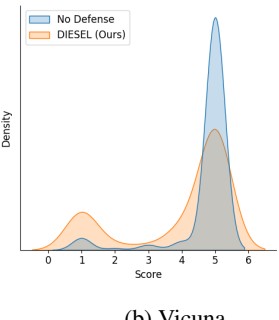

(a) Mistral  (b) Vicuna

Figure 5: LLM judge scores distribution when no defense and DIESEL are applied to different LLMs in a jailbreak scenario (GCG attack Zou et al. (2023))

### 4.2.3 Truthfulness & Coherency

Since DIESEL modifies the original token distribution generated by the LLM, we further investigate its impact on responses to benign (safe) prompts. For this evaluation, we used the TruthfulQA dataset and generated responses using Llama 3. Similar to the safety evaluation, we asked the LLM judge whether the produced response matched any of the ground-truth answers provided in the dataset. If the response did not align with the ground-truth answers, we further assessed whether it was semantically coherent or incoherent. From the results on Llama 3, we observe that as $\alpha$ increased to a certain threshold, the coherence of responses remained largely unaffected ($<5\%$) while maintaining comparable levels of truthfulness (51%) compared to the vanilla auto-regressive inference (60%). We include the evaluation prompt and detailed results in the supplementary material.

### 4.2.4 Inference Time Analysis

A key consideration for inference guidance techniques is the additional execution time they introduce. Table 1 compares the inference times for DIESEL and RAIN against a vanilla auto-regressive inference. For example, generating responses with Llama 3 using DIESEL results in only a 1.46× increase in runtime, which remains feasible for real-time applications. In contrast, RAIN introduces a prohibitive overhead, increasing

|  | Llama 3 | Mistral | Vicuna |
|---|---|---|---|
| RAIN | ×189.74 | × 186.45 | × 202.18 |
| DIESEL (Ours) | ×1.46 | ×1.52 | × 1.64 |

Table 1: Inference time comparison between RAIN and DIESEL. Values represent the inference time increase compared to a vanilla model.

runtime by 189× compared to the vanilla auto-regressive inference, rendering it unsuitable for practical use cases. We hypothesize that this drastic overhead arises from the use of conversational models in our evaluation, whereas RAIN's original results were based on non-chat models. Non-chat models typically produce shorter, more concise responses, whereas conversational models—fine-tuned to provide helpful and informative answers—tend to generate longer responses, significantly impacting RAIN's runtime. Importantly, as model size increases, the relative increase in execution time from using DIESEL becomes less pronounced compared to the model's base forward runtime.

### 4.2.5 User Study

To verify DIESEL's ability to generate safer responses compared to vanilla auto-regressive inference from a human perspective, we conducted a user study with 20 human evaluators, equally split between male and female participants. For each prompt, participants were given two responses and asked to rank each prompt on a scale of 1 to 5. A score of 1 indicates that "response 1" is substantially safer than "response 2" (vice versa for a score of 5), a score of 2 indicates that "response 1" is slightly safer than "response 2" (vice versa for a score of 4), and a score of 3 indicates that both responses are equally safe/unsafe. To reduce bias, the responses of DIESEL and vanilla auto-regressive inference were randomly shuffled for each participant and each prompt. The averaged results show that for 80% of DIESEL's responses are safer than those of a vanilla auto-regressive inference, 10% are equally safe, and the remaining 10% are slightly unsafer.

### 4.2.6 BEYOND SAFETY

To demonstrate the generalizability of DIESEL beyond safety-focused tasks, we conducted an experiment in the domain of movies. For this, we used the Wiki Movie Plots dataset Priya (2024), which provides detailed movie summaries across various genres. In this experiment, we tasked the model with summarizing plots of horror films while treating the genre "horror" as a negative concept. Specifically, we focus on horror movies, where the objective was to evaluate whether DIESEL could effectively reduce the presence of horror-related content in the generated summaries. Then, using the LLM judge, we compare the generated summaries with the originals from the dataset, asking the model to assess which summary exhibited more horror elements. The results indicated that 82% of the summaries generated with DIESEL contained fewer horror elements than their original counterparts, underscoring DIESEL's ability to suppress undesired content in domains beyond safety-related contexts.

## 5 LIMITATIONS

One limitation of DIESEL is that as the response length increases (during generation), the sentence embedding model faces challenges in accurately assessing the similarity between the generated content and predefined negative concepts. However, we observed that once the general "direction" of the response is established, subsequent tokens tend to follow the same trajectory, minimizing the impact of this limitation on the overall safety of the generated output. Future research could explore embedding models that are more specialized in handling long sequences, or dynamic strategies that adapt to the response length and complexity of generated responses. Another limitation of DIESEL relates to the irrevocable nature of token selection during each iteration. Once a token is selected at the end of an iteration, it cannot be undone. In some instances, a token chosen at iteration $i$ may not be flagged as unsafe in isolation but, when combined with a token selected in a subsequent iteration, may result in an unsafe sentence. While this issue could potentially be mitigated by employing a look-ahead mechanism, this approach would come at the cost of increased runtime. Given the trade-off between computational efficiency and safety, we opted to maintain a lightweight approach suitable for real-world applications.

## 6 CONCLUSION

In this paper, we introduced DIESEL, a novel lightweight inference guidance technique designed to enhance the safety of responses generated by large language models. We demonstrated that DIESEL effectively mitigates harmful outputs while maintaining coherence and relevance. Moreover, our evaluations against a competitive state-of-the-art inference guidance technique highlighted DIESEL 's practical advantages, including significantly lower runtime overhead, making it suitable for real-world applications. We also evaluated DIESEL's robustness against jailbreak attacks, showing that it offers an additional layer of defense even in adversarial contexts. Importantly, our method's reliance on simple textual descriptions of negative concepts allows it to be flexible, easily updated, and usable without specialized expertise. Finally, through both automated metrics and human evaluations, we verified DIESEL's ability to produce safer responses without compromising the quality or truthfulness of benign outputs.

## 7 ETHICAL IMPACT

This paper aims to enhance the safety of large language models (LLMs) by introducing a novel lightweight inference guidance technique. As LLMs find broader application in real-world scenarios, ensuring their safety becomes increasingly crucial. Importantly, the development of DIESEL does not involve crafting new jailbreak attacks but instead makes use of those that are already publicly available. For illustration, we include examples of harmful model responses. We acknowledge that the introduction of DIESEL may inspire the creation of new attack strategies aimed at circumventing its defenses. We will release the associated code and demonstrations to aid future red-teaming efforts in preventing LLM misuse.

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
