# OpenReview forum: "DIESEL - Dynamic Inference-Guidance via Evasion of Semantic Embeddings in LLMs"
_ICLR.cc/2025/Conference — ICLR 2025 Conference Withdrawn Submission_

### Official Review · Reviewer_sAns · 2024-11-01

**Soundness:** 2
**Presentation:** 3
**Contribution:** 2
**Rating:** 3
**Confidence:** 3

**Summary:**

This paper describes an inference time method to reduce the risk of generating harmful content. At each time step of decoding, it computes the cosine similarity between the set of harmful concepts and the currently generated output tokens + each candidate (from a  ranked list of top-k candidates) using a light-weight sentence embedder.  These scores are used to readjust the generation probabilities for the tokens. The method has been compared against RAIN (another decoding time method) and applied on 3 LLMs (Llama-3-8B, Vicuna-7B, and Mistral-7B) using the AdvBench dataset and TruthfulQA dataset. Automated evaluation for harmfulness (a GPT4 judge that rates whether a response is harmful or not) suggests that this method does better than RAIN and an evaluation on TruthfulQA using Llama 3(again using an automated judge) suggests a 9 percentage point absolute drop in performance.

Overall, this is a well-written easy to read paper. The algorithm blocks use variable names that are not defined and do not match the main text equations (eg: Line 8 in Algo Block 2 refers to $r_e$ and its missing the encoder (unless it refers to the encoded representation of a concept).

**Strengths:**

- Simple light-weight method
- Demonstrated on multiple models

**Weaknesses:**

- Evaluation is limited  (automated judges, no insight on failures, a single dataset)
- Noteably absent is discussion and comparison with work based on representation steering (eg: https://openreview.net/pdf?id=dJTChKgv3a)
- Limited comparison of models (eg: Does the technique work better or worse at different model sizes)

**Questions:**

Q1. As the authors mention, the sentence embedder becomes a bottleneck when the responses start to become more complex. Presumably this could be true for the harmfulness concepts set $R$ as well. There were no examples of the concepts in the appendix or the main paper. Are these sentences? Words?
Q2. Could this method help improve instruction-following (eg: In IFEval) by encoding instructions as concepts (instead of harms)?

---

### Official Review · Reviewer_Q1n4 · 2024-11-02

**Soundness:** 2
**Presentation:** 2
**Contribution:** 3
**Rating:** 3
**Confidence:** 3

**Summary:**

This paper introduces DIESEL, a lightweight inference guidance technique designed to act as a safeguard for LLMs. DIESEL reranks tokens proposed by the LLM based on their similarity to predefined negative concepts in latent space, using a sentence embedding model (miniLM). Experimental results indicate that, compared to RAIN, DIESEL is faster and provides enhanced safety.

**Strengths:**

Focuses on a critical area—LLM safety.

Lightweight and efficient reranker technique that proves effective on AdvBench over three LLM backbone (Llama 3, Mistral and Vicuna).

**Weaknesses:**

Limited Adversarial Benchmark:

Evaluation on only one adversarial benchmark with 500 prompts is insufficient to assess robustness. Since DIESEL relies on predefined unsafe terms, a small test set may not reveal limitations in generalizability. Testing on larger and more diverse adversarial datasets would provide a clearer measure of DIESEL’s robustness across varied unsafe cases.

Impact on General LLM Performance:

The authors need to confirm that DIESEL does not degrade performance on standard tasks. While results are provided on TruthfulQA, additional performance metrics on broader benchmarks like MMLU are necessary to evaluate the technique’s compatibility with general LLM tasks.

Evaluation Methodology:

Using an LLM judge (GPT4o) to assess safety raises concerns, as this may inherit limitations from the LLM itself. A more robust evaluation method, potentially involving human judges or external metrics, would better demonstrate DIESEL’s effectiveness.

**Questions:**

While reviewing your code, I’m trying to locate the predefined unsafe terms. Could you clarify what these terms are and where they are defined or stored?

---

### Official Review · Reviewer_f8De · 2024-11-04

**Soundness:** 3
**Presentation:** 3
**Contribution:** 2
**Rating:** 3
**Confidence:** 4

**Summary:**

This paper presents DIESEL, a new method to make language models generate safer responses without needing retraining. Think of it as a real-time safety filter that checks if what the AI is about to say is similar to any "bad" topics before letting it speak. The authors claim that this method only takes about 1.5 times longer than normal) and can be added to any AI chatbot. The authors tested it on several LLMs and showed it worked well at reducing harmful content while still letting the AI give helpful answers.

**Strengths:**

-  Originality: The paper introduces a **novel** approach using semantic embeddings for real-time safety filtering. This is the first paper I have ever seen to use semantic similarity to judge the safety of LLM outputs. This reminds me of the semantic uncertainty paper, which estimates the model hallucinations with the semantic similarity of the model outputs.
- The approach of using negative concepts is innovative and user-friendly. If you already have some bad cases in your mind, the proposed method could be easily used for it.
- Comprehensive evaluation across multiple models (Llama 3, Mistral, Vicuna), while it is a bit strange to use the Vicuna.
- The authors provide strong empirical validation with both automated metrics and human evaluation, clear ablation studies and runtime analysis, and testing against jailbreak attacks.
- The proposed method only requires minimal computational overhead (only 1.4-1.6x slower). It can easily be integrated with existing safety measures

**Weaknesses:**

- Limited Generalizability: Does not scale well to long-form output. No clear pathway for extending to longer text. Should include quantitative analysis showing how performance degrades with output length.
- Even for the short form of generations, the accuracy of the semantical similarity is questionable: The core assumption that semantic similarity can identify undesired content lacks rigorous validation.
- Semantic meaning isn't always indicative of toxicity. Many problematic outputs (like information leakage, implicit bias) may not have obvious semantic markers. For example, Information leakage examples show harmful content may not be semantically toxic. The paper should provide controlled experiments demonstrating the relationship between semantic similarity and different types of unsafe content.
- In addition, this method requires comparison with the pre-defined negative concepts R. What is the coverage of your pre-defined negative concepts? How could this be scaled up? It is not feasible to cover all possible negative responses. This is expensive and not scalable.
- The proposed method introduces too many additional hyper-parameters to tune, such as (b=20 tokens, α=0.98). It might be helpful to understand the sensitivity of the proposed method to these values.
- Limited comparison to SOTA methods like Llama Guard

**Questions:**

N/A

---

### Official Review · Reviewer_Zw2t · 2024-11-06

**Soundness:** 3
**Presentation:** 3
**Contribution:** 2
**Rating:** 3
**Confidence:** 3

**Summary:**

This paper presents DIESEL, a technique designed to improve the safety of LLMs by reranking tokens during inference based on their similarity to predefined negative concepts. DIESEL operates efficiently with minimal resources and does not require fine-tuning the model. The method involves three steps: candidate selection, semantic latent space similarity, and token reranking using an off-the-shelf sentence embedding model. The paper evaluates DIESEL across three open-source LLMs, assessing its performance both as a standalone safeguard and as an additional layer of defense, as well as its robustness against jailbreaking attacks.

**Strengths:**

1. The paper studies an interesting problem and is clearly structured.

2. The paper introduces an effective method for improving the safety of conversational LLMs.

**Weaknesses:**

1. The use of similarity-based decoding control to improve the safety of generation seems to be a very common idea.

2. Only one baseline is compared, and there is a lack of detailed description of the baseline model.

3. In terms of safety evaluation, the author only conducts experiments on one dataset. It would be better to refer to [1] and perform experiments on more datasets, thus making the experiment more convincing.

[1] Dong Z, Zhou Z, Yang C, et al. Attacks, defenses and evaluations for llm conversation safety: A survey[J]. arXiv preprint arXiv:2402.09283, 2024.

**Questions:**

n/a

---

### Note · Authors · 2024-12-04

I have read and agree with the venue's withdrawal policy on behalf of myself and my co-authors.